# Impact of the COVID-19 Pandemic on In-Patient Treatment of Peripheral Artery Disease in Germany during the First Pandemic Wave

**DOI:** 10.3390/jcm11072008

**Published:** 2022-04-03

**Authors:** Christian Scheurig-Muenkler, Florian Schwarz, Thomas J. Kroencke, Josua A. Decker

**Affiliations:** Department of Diagnostic and Interventional Radiology, University Hospital Augsburg, 86156 Augsburg, Germany; christian.scheurig@uk-augsburg.de (C.S.-M.); florian.schwarz@uk-augsburg.de (F.S.); josua.decker@uk-augsburg.de (J.A.D.)

**Keywords:** COVID-19 pandemic, public health, peripheral artery disease, in-hospital outcomes

## Abstract

Patients with peripheral artery disease (PAD) belong to a vulnerable population with relevant comorbidity. Appropriate care and timely treatment are imperative, but not readily assured in the current pandemic. What impact did the first wave have on in-hospital treatment in Germany? Nationwide healthcare remuneration data for inpatient care of the years 2019 and 2020 were used to compare demographic baseline data including the assessment of comorbidity (van Walraven score), as well as the encoded treatments. A direct comparison was made between the first wave of infections in 2020 and the reference period in 2019. The number of inpatient admissions decreased by 10.9%, with a relative increase in hospitalizations due to PAD Fontaine IV (+13.6%). Baseline demographics and comorbidity showed no relevant differences. The proportion of emergency admissions increased from 23.4% to 28.3% during the first wave to the reference period in 2019, and in-hospital mortality increased by 21.9% from 2.5% to 3.1%. Minor and major amputations increased by 24.5% and 18.5%. Endovascular and combined surgical/endovascular treatment strategies increased for all stages. Already in the first, comparatively mild wave of the pandemic, significantly fewer patients with predominantly higher-grade PAD stages were treated as inpatients. Consecutively, in-hospital mortality and amputation rates increased.

## 1. Introduction

The number of hospitalizations of patients suffering from peripheral artery disease (PAD) has steadily increased over the past decades, with a clear trend towards endovascular therapy [1,2]. An increase in the treatment of older and more morbid patients with a corresponding increase in nursing care was also observed [2]. In view of the demographic change, this trend is expected to continue. However, the global SARS-CoV-2 pandemic suddenly presented tremendous challenges to even the most efficient health care systems. To create capacity to care for COVID-19 patients, non-acutely necessary treatments were postponed in all health care areas. For stroke and myocardial infarction care, the negative impact of the pandemic on acute care with delayed treatments has already been documented [3,4,5,6,7,8,9]. In PAD patients, it was considered safer to manage claudicants conservatively and to hospitalize only if relevant deterioration occurred [10]. Even patients with chronic but non-critical wounds were not necessarily treated directly at the peak of each pandemic wave. In North America, treatment numbers dropped by 35.2% in the course of the first pandemic wave [11]. In addition, there was the uncertainty and fear of the patients themselves, which is why they often avoided going to the doctor or even to hospital in order not to expose themselves, supposedly unnecessarily, to the risk of infection. This is also reflected in the fact that the adequate and thus vital control of risk factors has relevantly deteriorated [12]. But how did the 2020 pandemic, particularly the first surge from mid-February to mid-May, ultimately affect inpatient treatment of PAD patients in Germany compared to the previous year? Here, it is important to add that in Germany, unlike in many other countries, revascularizations are almost exclusively performed on an inpatient basis and patients as well as referring/treating physicians have hardly any alternatives available to them. To demonstrate the acute impact of the pandemic, a nationwide analysis was conducted of all patients hospitalized primarily for treatment of PAD in 2019 and 2020.

## 2. Materials and Methods

### 2.1. Data Source

Access to the nationwide inpatient data set, summarizing all anonymized inpatient treatment data for accounting purposes in German hospitals, was provided by the Research Data Center (RDC) of the German Federal Statistical Office [13]. We analyzed demographic characteristics, diagnoses, procedures, remuneration, and in-hospital outcomes of all patients admitted for symptomatic PAD in the years 2019 and 2020. In addition, we analyzed the period of the first pandemic wave from mid-February to mid-May 2020 compared to the 2019 reference period. Using dedicated syntaxes written by the authors, analyses were remotely conducted, and the results were transferred after passing an anonymity check. Subgroups of fewer than 5 individual cases were censored.

### 2.2. German Diagnosis Related Groups (G-DRG) Remuneration System

Based on the main (reason for admission) and secondary diagnoses (comorbidities) as well as all procedures conducted during hospitalization, each individual case was assigned to a case-specific DRG, which determines the corresponding lump-sum remuneration. All diagnoses were encoded using the International Classification of Diseases 10th Revision in its German modification (ICD-10-GM). Procedures performed during hospitalization were coded using the Operation and Procedure Classification System (OPS).

### 2.3. Patient Cohort

All completed inpatient treatment cases in 2019 and 2020 with symptomatic PAD, corresponding to a Fontaine stage IIb or higher as the main diagnosis were included. In addition to the number of cases, age, gender, and length of stay, minor and major amputation rates and in-hospital mortality were determined as outcome measures for both years comparatively overall and subdivided by the Fontaine stage. Secondary diagnoses were used to calculate the weighted linear van Walraven score (vWs) to determine the average comorbidity of the cohort studied. Based on the individually encoded OPS codes, 3 treatment categories were defined: (1) percutaneous endovascular revascularization, including all sole endovascular procedures such as angioplasty, stents and atherectomy; (2) surgical revascularization, including procedures such as open thrombectomies, endarterectomies and bypass surgery; and (3) combined revascularization, in which endovascular and surgical procedures were combined in a one-stage hybrid procedure or performed separately in two steps.

### 2.4. Statistical Analysis

Data analysis and coding for controlled remote data processing were performed using R version 4.1.0 (https://www.r-project.org/, accessed on 10 December 2020). Calculation of the weighted linear van Walraven score was performed using the R package comorbidity (https://cran.r-project.org/package=comorbidity, accessed on 10 December 2020) [14]. Continuous variables were presented as mean with standard deviation (SD). Absolute and relative changes were given in numbers and percentages.

## 3. Results

In 2020, the first year of the COVID-19 pandemic in Germany, the total number of hospitalizations of PAD patients decreased by 10.9% from 185,713 to 165,554, with the greatest decline in claudicants (−17.8%) and patients with chronic wounds (−8.3%). Hospitalizations decreased less for patients with pain at rest (−4.3%) and gangrene (−1.4%). Looking only at the period of the first pandemic wave from mid-February to mid-May, the observed effect was even more pronounced. Inpatient admissions for symptomatic PAD decreased by 22.5% from 46,546 in 2019 to 36,096 in 2020. Table 1 summarizes the corresponding detailed data comparatively for the entire years 2019 and 2020, and Table 2 summarizes those for the first pandemic surge in 2020 compared with the same period in 2019. During the same period, the mean percentage of admissions as emergency compared to elective increased from 23.4% to 28.3%, which was comparable to the ratios during the Christmas and New Year period. Figure 1 illustrates the effect of the first wave of the pandemic, with a substantial decrease in hospital admissions from mid-February to mid-May and a corresponding relative increase in emergency admissions. No relevant difference was found between the two years with regard to general demographic data such as age and gender and comorbidity.

In terms of in-hospital outcome, there has been a substantial increase in rates of minor and major amputations of 14.5% and 8.6%, respectively, and in-hospital mortality with an increase of 11.3% across all stages, the latter two being of particular concern (Table 3 and Table 4). Looking only at the first wave period, the difference was much more severe, with a 21.9% increase in in-hospital mortality from 2.5% to 3.1%. Minor and major amputations increased by 24.5% and 18.5%, respectively, during this period.

A small percentage of patients (585/165,554; 0.35%) admitted for treatment of PAD in 2020 had COVID-19 infection coded as a secondary diagnosis. The mortality rate in this patient population was 25.7% compared with 2.6% in patients without concomitant COVID-19 infection. In the presence of PAD Fontaine IV, the mortality rate was as high as 30% compared with only 5.3% in patients without infection.

In addition to pandemic-related patient selection, a change in the chosen treatment modality was also observed. The number of hospitalizations without revascularization decreased across all stages. Endovascular recanalization, but more clearly, combined revascularizations with hybrid- or two-step surgical and endovascular treatments increased considerably (Table 5).

## 4. Discussion

At first glance, the pandemic has upended trends in in-hospital outcomes over the past decades, and the relative increase in amputations and in-hospital mortality may baffle [1,2]. But is this development due to systematically worse patient care? Presumably, pandemic-related patient selection is to blame for this in no small part. While in 2019, claudicants accounted for 46.9% of all hospitalizations, in 2020 the share was only 43.2%. This shift in favor of patients with higher-grade PAD naturally also affects the immediate outcome. However, analysis by stage also shows an increase in mortality in patients with PAD Fontaine IV. At the same time, the number of minor amputations is also increasing among them. It should be noted that the Fontaine stage classification used for the mixed calculation of DRG remuneration is very imprecise and that, for example patients in stage IVu alone, i.e., patients with chronic ulcerations and still without necrosis, may vary greatly in severity without this being coded. Thus, the absolute treatment numbers of patients with PAD Fontaine IV have also decreased, although presumably heterogeneously across the subgroup. Using data from a North American quality assurance registry (Vascular Quality Initiative), Lou et al. also found a decrease in elective and especially low-grade PAD treatments [11]. However, they were also able to demonstrate that the complexity of interventional cases has increased considerably. Thus, the number of patients with TASC-D lesions in the first wave increased from 30.1% to 35.3%, the number of crural interventions from 34.3% to 40.7%, and the mean length of treated occlusions from 8.7 ± 16.7 cm to 11.0 ± 23.1 cm. It is plausible that such outcome-relevant shifts have also occurred in Germany. In a retrospective single-center study in Switzerland, Trunfio et al. demonstrated a similar effect, with an overall decrease in patient numbers but a relative increase in Fontaine stages III-IV to 66.7% during the first wave compared with 47.3% during the same periods in 2018 and 2019. Equally impressive was the relative increase in the proportion of patients with acute limb ischemia, i.e., a vascular emergency, from 24.6% to 47.5% [12]. In contrast, in their evaluation of two large hospitals in the Netherlands, Exelmans et al. observed no significant decrease in the number of patients treated throughout 2020, but a relative increase in higher PAD stages and acute limb ischemia with a subsequent increase in major amputations [15]. Thus, care for critically ill patients does not seem to be that compromised and the outcome acceptable? On the contrary, these observations merely show the tip of the iceberg and are an indication that many of these vulnerably ill patients are not being adequately cared for and treated [16,17]. Waiting until chronic ischemia becomes critical, ischemia can mean the difference not only between amputation yes or no, but also between life and death. Changes in the already high comorbidity during the first wave could not be observed. Concomitant COVID-19 infection also did not play a critical role, with a prevalence of only 0.35%, although mortality and amputation rates were likely higher in this subgroup. However, it must be mentioned that only patients whose main admission diagnosis was PAD were included. COVID-19 infection was coded only as a secondary diagnosis and either existed at admission or occurred during the stay. However, if the course of such an infection is a severe and hospitalization-dominant, the original main diagnosis may well have been changed. Therapies and outcomes for secondary diagnoses of PAD were not included in this evaluation. In addition to an acute infection, however, adverse effects can also harm these patients in the long term after an infection has long been passed through. Asarcikli et al. described a dysregulation of the autonomic nervous system after COVID infection as a possible cause for the circulatory problems in patients with post-COVID syndrome [18]. Such imbalances affect patients with already impaired cardiovascular function even more. Of course, these effects cannot be assessed on the basis of the evaluated data.

The observed changes in treatment choice may well be a consequence of changes in patient selection but may also be due to changes in the way pandemic-related resource constraints are managed. Even though the impact on the comparatively high bed capacity of intensive care units in Germany in the first wave was lower than in other European countries, this could only be achieved through a consistent lockdown and a resource-saving approach [19]. Without this, the capacities would also have been quickly exhausted [20]. With this awareness, supposedly unnecessary operations with necessary follow-up in the intensive care unit were postponed or treated otherwise, e.g., interventional.

The structural limitations of this evaluation are obvious. Quality of treatment can only be fully assessed through longitudinal analysis of individual patient outcomes, including outpatient care data. It is also conceivable that some of the increased numbers of interventions were only provisional in nature and that the final treatment was to take place during another, later hospitalization. A corresponding systematic effect on treatment numbers could not be derived from the data, since readmissions cannot be filtered out. It remains a snapshot, but impressively reflects the changed demand during the first wave of the pandemic. However, only the first wave. By 31.12.2020, a good 1.7 million people in Germany were infected with COVID-19. This means that the first wave hit Germany comparatively mildly. By the following year, 2021, nearly 7 million people were already cumulatively infected. The corresponding impact of the more dramatic surges in 2021 has yet to be assessed.

## 5. Conclusions

In summary, in the first wave of the pandemic, the selection of patients requiring inpatient care for their PAD changed relevantly towards higher stages with regard to the severity of their disease. Patients were increasingly treated with endovascular or combined surgical/endovascular treatment. Presumably as a result of patient selection, an increase in in-hospital mortality and amputation rates was observed. However, these observations reflect only the effects of the first, mildest of all pandemic waves.

## Figures and Tables

**Figure 1 jcm-11-02008-f001:**
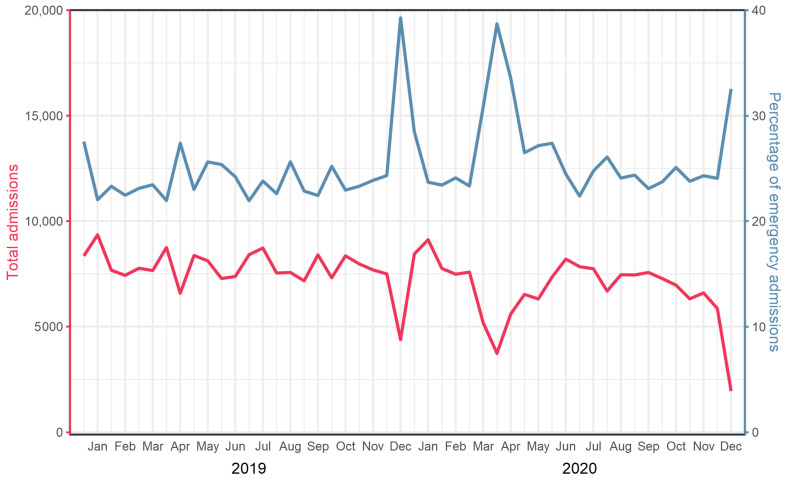
Semi-monthly numbers of total PAD admissions (red line) and percentage of emergency admissions (blue line) in Germany in 2019 and 2020.

**Table 1 jcm-11-02008-t001:** Baseline characteristics of patients hospitalized due to peripheral artery disease in 2019 and 2020.

	Fontaine	2019	2020	Absolute Change	Relative Change
Hospitalizations	total	185,713 (100.0)	165,554 (100.0)	−20,159 (−10.9%)	
	IIb	87,067 (46.9)	71,529 (43.2)	−15,538 (−17.8%)	−7.8%
	III	22,882 (12.3)	21,893 (13.2)	−989 (−4.3%)	+7.3%
	IVu	37,722 (20.3)	34,607 (20.9)	−3115 (−8.3%)	+2.9%
	IVg	38,042 (20.5)	37,525 (22.7)	−517 (−1.4%)	+10.7%
Hospitalizations per 100,000 inhabitants	total	223.3	199.1	−24.2 (−10.8%)	
	IIb	104.7	86.0	−18.7 (−17.8%)	
	III	27.5	26.3	−1.2 (−4.3%)	
	IVu	45.4	41.6	−3.7 (−8.2%)	
	IVg	45.7	45.1	−0.6 (−1.3%)	
Sex (male)	total	117,576 (63.3)	104,716 (63.3)	−12,860 (−10.9%)	−0.1%
	IIb	57,420 (65.9)	46,791 (65.4)	−10,629 (−18.5%)	−0.8%
	III	13,315 (58.2)	12,617 (57.6)	−698 (−5.2%)	−1.0%
	IVu	21,887 (58.0)	20,497 (59.2)	−1390 (−6.4%)	+2.1%
	IVg	24,954 (65.6)	24,811 (66.1)	−143 (−0.6%)	+0.8%
Age (years)	total	72.0 ± 10.9	72.2 ± 10.9	+0.2	
	IIb	68.4 ± 9.8	68.4 ± 9.8	0.0	
	III	71.6 ± 10.9	71.4 ± 10.8	−0.2	
	IVu	76.7 ± 10.5	76.5 ± 10.4	−0.1	
	IVg	76.0 ± 10.8	75.8 ± 10.8	−0.2	
In-hospital stay (days)	total	9.2 ± 12.1	9.2 ± 12.0	0.0	
	IIb	4.3 ± 5.4	4.1 ± 5.4	−0.2	
	III	9.3 ± 10.0	8.9 ± 10.1	−0.4	
	IVu	11.8 ± 12.4	11.2 ± 11.9	−0.6	
	IVg	17.9 ± 17.1	17.2 ± 16.5	−0.6	
van Walraven score	total	7.0 ± 6.6	7.2 ± 6.7	+0.1	
	IIb	4.6 ± 4.7	4.6 ± 4.6	0.0	
	III	6.9 ± 6.4	6.8 ± 6.2	−0.1	
	IVu	9.1 ± 7.2	9.1 ± 7.1	−0.1	
	IVg	10.6 ± 7.3	10.5 ± 7.6	0.0	

Data are number (percentage) or mean ± standard deviation. IIb = Fontaine IIb; III = Fontaine III; IVu = Fontaine IV with ulcers; IVg = Fontaine IV with gangrene.

**Table 2 jcm-11-02008-t002:** Baseline characteristics of patients hospitalized due to peripheral artery disease from mid-February to mid-May in 2019 and 2020.

	Fontaine	16 February 2019 to 15 May 2019	16 February 2020 to 15 May 2020	Absolute Change	Relative Change
Hospitalizations	total	46,546 (100.0)	36,096 (100.0)	−10,450 (−22.5%)	
	IIb	21,503 (46.2)	13,738 (38.1)	−7765 (−36.1%)	−17.6%
	III	5720 (12.3)	5230 (14.5)	−490 (−8.6%)	+17.9%
	IVu	9449 (20.3)	7924 (22.0)	−1525 (−16.1%)	+8.1%
	IVg	9874 (21.2)	9204 (25.5)	−670 (−6.8%)	+20.2%
Hospitalizations per 100,000 inhabitants	total	56.0	43.4	−12.6 (−22.4%)	
	IIb	25.9	16.5	−9.3 (−36.1%)	
	III	6.9	6.3	−0.6 (−8.6%)	
	IVu	11.4	9.5	−1.8 (−16.1%)	
	IVg	11.9	11.1	−0.8 (−6.8%)	
Sex (male)	total	29,531 (63.4)	22,999 (63.7)	−6532 (−22.1%)	+0.4%
	IIb	14,205 (66.1)	9124 (66.4)	−5081 (−35.8%)	+0.5%
	III	3360 (58.7)	3035 (58.0)	−325 (−9.7%)	−1.2%
	IVu	5518 (58.4)	4756 (60.0)	−762 (−13.8%)	+2.8%
	IVg	6448 (65.3)	6084 (66.1)	−364 (−5.6%)	+1.2%
Age (years)	total	71.9 ± 11.0	72.3 ± 11.0	+0.3	
	IIb	68.1 ± 9.9	68.2 ± 9.8	0.0	
	III	71.6 ± 10.9	70.7 ± 10.8	−0.8	
	IVu	76.7 ± 10.4	76.4 ± 10.3	−0.3	
	IVg	75.9 ± 10.9	75.8 ± 10.7	−0.2	
In-hospital stay (days)	total	9.3 ± 12.2	9.4 ± 11.9	+0.1	
	IIb	4.3 ± 5.1	4.0 ± 5.2	−0.2	
	III	9.5 ± 10.5	8.6 ± 9.6	−0.9	
	IVu	11.6 ± 12.6	10.8 ± 11.2	−0.8	
	IVg	17.7 ± 17.1	16.7 ± 16.1	−1.1	
van Walraven score	total	7.0 ± 6.6	7.3 ± 6.8	+0.3	
	IIb	4.5 ± 4.7	4.5 ± 4.6	0.0	
	III	6.9 ± 6.5	6.8 ± 6.3	−0.1	
	IVu	9.0 ± 7.2	8.9 ± 7.1	−0.2	
	IVg	10.5 ± 7.6	10.5 ± 7.6	0.0	

Data are number (percentage) or mean ± standard deviation. IIb = Fontaine IIb; III = Fontaine III; IVu = Fontaine IV with ulcers; IVg = Fontaine IV with gangrene.

**Table 3 jcm-11-02008-t003:** In-hospital outcome of patients admitted due to peripheral artery disease in 2019 and 2020.

	Fontaine	2019	2020	Absolute Change	Relative Change
Major amputation	total	6436 (3.5)	6228 (3.8)	−208 (−3.2%)	+8.6%
	IIb	17 (0.0)	17 (0.0)	0 (0.0%)	+21.7%
	III	308 (1.3)	283 (1.3)	−25 (−8.1%)	−4.0%
	IVu	858 (2.3)	830 (2.4)	−28 (−3.3%)	+5.4%
	IVg	5253 (13.8)	5098 (13.6)	−155 (−3.0%)	−1.6%
Minor amputation	total	15,080 (8.1)	15,394 (9.3)	+314 (+2.1%)	+14.5%
	IIb	33 (0.0)	20 (0.0)	−13 (−39.4%)	−26.2%
	III	61 (0.3)	56 (0.3)	−5 (−8.2%)	−4.0%
	IVu	2779 (7.4)	2845 (8.2)	+66 (+2.4%)	+11.6%
	IVg	12,207 (32.1)	12,473 (33.2)	+266 (+2.2%)	+3.6%
In-hospital mortality	total	4600 (2.5)	4562 (2.8)	−38 (−0.8%)	+11.3%
	IIb	180 (0.2)	147 (0.2)	−33 (−18.3%)	−0.6%
	III	462 (2.0)	414 (1.9)	−48 (−10.4%)	−6.3%
	IVu	1203 (3.2)	1168 (3.4)	−35 (−2.9%)	+5.8%
	IVg	2755 (7.2)	2833 (7.5)	+78 (+2.8%)	+4.2%

Data are number (percentage). IIb = Fontaine IIb; III = Fontaine III; IVu = Fontaine IV with ulcers; IVg = Fontaine IV with gangrene.

**Table 4 jcm-11-02008-t004:** In-hospital outcome of patients admitted due to peripheral artery disease from mid-February to mid-May in 2019 and 2020.

	Fontaine	16 February 2019 to 15 May 2019	16 February 2020 to 15 May 2020	Absolute Change	Relative Change
Major amputation	total	1599 (3.4)	1470 (4.1)	−129 (−8.1%)	+18.5%
	IIb	0 (0.0)	0 (0.0)	0 (0.0%)	0.0%
	III	91 (1.6)	77 (1.5)	−14 (−15.4%)	−7.5%
	IVu	216 (2.3)	192 (2.4)	−24 (−11.1%)	+6.0%
	IVg	1292 (13.1)	1201 (13.0)	−91 (−7.0%)	−0.3%
Minor amputation	total	3894 (8.4)	3759 (10.4)	−135 (−3.5%)	+24.5%
	IIb	6 (0.0)	6 (0.0)	0 (0.0%)	+56.5%
	III	19 (0.3)	9 (0.2)	−10 (−52.6%)	−48.2%
	IVu	708 (7.5)	676 (8.5)	−32 (−4.5%)	+13.9%
	IVg	3161 (32.0)	3068 (33.3)	−93 (−2.9%)	+4.1%
In-hospital mortality	total	1171 (2.5)	1107 (3.1)	−64 (−5.5%)	+21.9%
	IIb	51 (0.2)	33 (0.2)	−18 (−35.3%)	+1.3%
	III	122 (2.1)	98 (1.9)	−24 (−19.7%)	−12.1%
	IVu	298 (3.2)	278 (3.5)	−20 (−6.7%)	+11.2%
	IVg	700 (7.1)	698 (7.6)	−2 (−0.3%)	+7.0%

Data are number (percentage). IIb = Fontaine IIb; III = Fontaine III; IVu = Fontaine IV with ulcers; IVg = Fontaine IV with gangrene.

**Table 5 jcm-11-02008-t005:** Type of treatment of patients hospitalized due to peripheral artery disease in 2019 and 2020.

Fontaine Stage	Treatment	2019	2020	Absolute Change	Relative Change
All	No intervention	47,287 (25.5)	40,492 (24.5)	−6795 (−14.4%)	−3.9%
	Endovascular	93,902 (50.6)	84,274 (50.9)	−9628 (−10.3%)	+0.7%
	Surgical	27,756 (14.9)	24,536 (14.8)	−3220 (−11.6%)	−0.8%
	Combined	16,768 (9.0)	16,252 (9.8)	−516 (−3.1%)	+8.7%
IIb	No intervention	10,704 (12.3)	7934 (11.1)	−2770 (−25.9%)	−9.8%
	Endovascular	55,799 (64.1)	46,705 (65.3)	−9094 (−16.3%)	+1.9%
	Surgical	12,996 (14.9)	10,388 (14.5)	−2608 (−20.1%)	−2.7%
	Combined	7568 (8.7)	6502 (9.1)	−1066 (−14.1%)	+4.6%
III	No intervention	4753 (20.8)	3925 (17.9)	−828 (−17.4%)	−13.7%
	Endovascular	9452 (41.3)	9298 (42.5)	−154 (−1.6%)	+2.8%
	Surgical	5227 (22.8)	5148 (23.5)	−79 (−1.5%)	+2.9%
	Combined	3450 (15.1)	3522 (16.1)	72 (2.1%)	+6.7%
IVu	No intervention	15,056 (39.9)	12,726 (36.8)	−2330 (−15.5%)	−7.9%
	Endovascular	15,992 (42.4)	15,445 (44.6)	−547 (−3.4%)	+5.3%
	Surgical	4300 (11.4)	3915 (11.3)	−385 (−9.0%)	−0.8%
	Combined	2374 (6.3)	2521 (7.3)	147 (6.2%)	+15.8%
IVg	No intervention	16,774 (44.1)	15,907 (42.4)	−867 (−5.2%)	−3.9%
	Endovascular	12,659 (33.3)	12,826 (34.2)	167 (1.3%)	+2.7%
	Surgical	5233 (13.8)	5085 (13.6)	−148 (−2.8%)	−1.5%
	Combined	3376 (8.9)	3707 (9.9)	331 (9.8%)	+11.3%

Data are number (percentage). IIb = Fontaine IIb; III = Fontaine III; IVu = Fontaine IV with ulcers; IVg = Fontaine IV with gangrene.

## Data Availability

The access to the analyzed data was provided by the Research Data Center (RDC) of the Federal Statistical Office and Statistical Offices of the Federal States, DRG Statistics 2019 and 2020. Using data structure files provided by the RDC, statistical syntaxes were written in R by the authors and sent to the RDC for processing the raw data. The results were subjected to a secrecy check censoring subgroups with fewer than five individuals to assure anonymity of individual cases. The results were then sent back to the authors.

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
