# Peer review of "Impact of the COVID-19 Pandemic on In-Patient Treatment of Peripheral Artery Disease in Germany during the First Pandemic Wave"

_jcm, 2022, doi:10.3390/jcm11072008_

Round 1

Reviewer 1 Report

The authors well describe an emergent problem happened during pandemic period, concerning others disease different from covid-19.

In particular they report the epidemiology during the pandemic period of PAD patient

A direct comparison was made between the first wave of infections in 2020 and the reference period in 2019. The number of inpatient admissions decreased by 10.9%, with a relative increase in hospitalizations due to PAD Fontaine IV (+13.6%). Baseline demographics and comorbidity showed no relevant differences. The proportion of emergency admissions increased from 23.4% to 28.3% during the first wave to the reference period in 2019, and in-hospital mortality increased by 21.9% from 2.5% to 3.1%. Minor and major amputations increased by 24.5% and 18.5%. Endovascular and combined surgical/endovascular treatment strategies increased for all stages. Already in the first, comparatively mild wave of the pandemic, significantly fewer patients with predominantly higher-grade PAD stages were treated as inpatients. Consecutively, in-hospital mortality and amputation rates increased.

These conclusions similar also for several "non covid disease" are very interesting for scientific community, and clearly confirmed the dramatic period also for  "non covid" patients.

Reviewer 2 Report

I have reviewed the manuscript entitled 'Impact of the COVID-19 pandemic on in-patient treatment of peripheral artery disease in Germany during the first pandemic wave'.

The manuscript is well-designed and written however several minor changes are required.

English grammar and typo errors should be corrected.

Similar delays are encountered in patients with STEMI, thus the effect of pandemic can be disastrous. The authors should mention this in the last section and consider citing 'Effect of the COVID-19 pandemic on access to primary percutaneous coronary intervention for ST-segment elevation myocardial infarction'.

The increased mortality can be due to concomitant covid-19 infection in these frail patients. Covid-19 can exacerbate several cardiac disease, this should also be mentioned in the discussion. The authors should consider citing newly released paper entitled 'Heart rate variability and cardiac autonomic functions in post-COVID period'.
